# Digital spatial profiling of the microenvironment of muscle invasive bladder cancer
Michael Eyers [1] ✉, Joely Irlam[2], Gayle Marshall [1], Vicky Smith [3], Alexander Baker [3], Lucy Frost[1], Peter Hoskin [2,4], Ananya Choudhury[2,5] & Catharine West [2]

Muscle invasive bladder cancer (MIBC) is a molecularly diverse disease with varied clinical outcomes. Molecular studies typically employ bulk sequencing analysis, giving a transcriptomic snapshot of a section of the tumour. However, tumour tissues are not homogeneous, but are composed of distinct compartments such as the tumour and stroma. To investigate the molecular profiles of bladder cancer, whilst also maintaining the spatial complexity of the tumours, we employed whole transcriptome Digital Spatial Profiling (DSP). With this method we generated a dataset of transcriptomic profiles of tumour epithelium, stroma, and immune infiltrate. With these data we investigate the spatial relationship of molecular subtype signatures and ligand signalling events. We find that Basal/Squamous and Classical subtypes are mostly restricted to tumour regions, while the stroma-rich subtype signatures are abundant within the stroma itself. Additionally, we identify ligand signalling events occurring between tumour, stroma, and immune infiltrate regions, such as immune infiltrate derived GPNMB, which was highly correlated with VEGFA expression within the tumour. These findings give us new insights into the diversity of MIBC at a molecular level and provide a dataset with detailed spatial information that was not available before in bladder cancer research.

Molecular analysis using bulk sequencing have revealed insights into the heterogeneity of bladder cancer, such as molecular signatures linked to patient outcomes and treatment response[1]. However, these techniques provide no information on spatial relationships between different compartments within a tissue, such as tumour and stroma, meaning that biologically significant differences may be lost due to homogenisation. Whole transcriptome Digital Spatial Profiling (DSP) is an emerging tool that has the potential to improve our understanding of bladder cancer heterogeneity by comprehensively mapping the spatial and morphological features of cancer[2].

There have been only a few spatial transcriptomics studies in bladder cancer. Initial unpublished results by Reeves et al.[3] showed molecular subtyping was consistent between traditional methods and DSP using an early panel profiling ~150 genes[3]. Studies of DSP in other cancers revealed novel insights into the tumour microenvironment to improve risk stratification[4] and identify interactions likely to affect biological processes[5].

Here we aimed to investigate the ability of DSP to spatially profile MIBC using tissue microarrays (TMAs) generated from archival FFPE samples. Our work shows the success of the method in archival bladder cancer samples. We show (1) a clear enrichment of tumour epithelium, stroma and immune infiltrated tumour cell gene signatures and cell types in their respective regions of interest; (2) good concordance between DSP and bulk sequencing classifications of basal/squamous and luminal molecular subtypes; and (3) the ability to identify key genes and pathways involved in molecular subtypes in situ. Finally, we developed a method to investigate ligand-signalling events between adjacent regions in DSP datasets which takes advantage of the spatial aspect of the data.

## Results

### Digital spatial transcriptomics of human bladder cancer TMAs

Figure 1a illustrates whole transcriptome spatial profiling of a TMA core, and Fig. 1b the distinction of tumour (Pan-CK+ve) and immune (CD45+ve) cells. A single region was selected within TMA cores and segmented into tumour (Pan-CK+ve) and stroma (Pan-CK-ve) ROIs. Figure 1c shows the identification of infiltrating immune cells (Pan-CK+ve and CD45+ve) in some cores.

[1]Medicines Discovery Catapult, Alderly Park, Cheshire, UK. [2]Division of Cancer Sciences, University of Manchester, Manchester, UK. [3]CRUK Manchester Centre, Manchester, UK. [4]Mount Vernon Centre for Cancer Treatment, Northwood, UK. [5]The Christie NHS Foundation Trust, Manchester, UK. ✉e-mail: Michael.eyres@md.catapult.org.uk

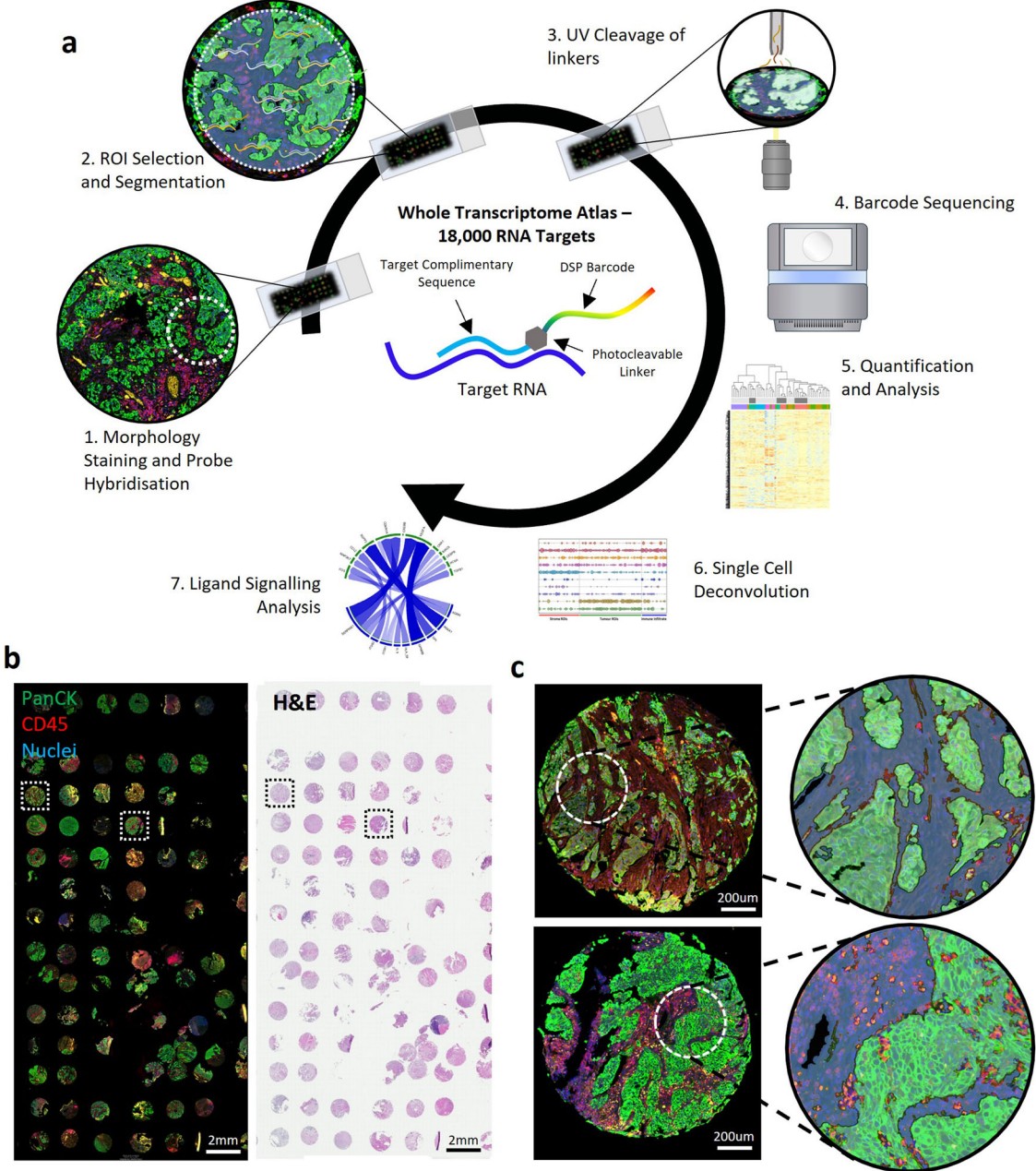

**Fig. 1 | Spatial transcriptomics of muscle invasive bladder cancer. a** Overview of the GeoMx DSP method. The Whole Transcriptome Atlas (WTA) panel consists of over 18,000 RNA probes attached to sequencing barcodes via a photocleavable linker. ROIs are selected and exposed to light to cleave all barcodes in the region. Sequencing of barcodes quantifies the RNA probes in each ROI to generate data for downstream analysis. **b** Bladder cancer TMAs from the BCON cohort stained with PanCK and CD45 (left) and haematoxylin & eosin. **c** TMA cores showing selection of ROIs. In each core a 300 μm diameter circle was selected and segmented using morphology marker staining for tumour, stroma, and immune infiltrated ROIs.

## Spatially resolved tumour regions have distinct gene expression profiles

There was a clear separation of the transcriptional profiles of tumour, immune infiltrated, and stroma ROIs by UMAP dimensional reduction (Fig. 2a). Tumour and immune infiltrated ROIs were enriched for epithelial markers such as *CDH1*, and stroma ROIs were enriched for genes associated with the extracellular matrix such as *FN1* and *ACTA2*. Immune infiltrated and stroma ROIs were also enriched for immune genes such as *CD68* (Fig. 2b). Differential expression, followed by IPA analysis showed that tumour ROIs were enriched for genes involved in translation elongation and EIF2 signalling, while stroma ROIs were enriched for collagen fibre reorganisation genes, such as *COL3A1*, and other ECM gene sets (Fig. 2c, Supplementary Fig. 1). Immune infiltrated ROIs were enriched for

neutrophil degranulation genes, such as *CXCL8*, as well as Class I MHC processing and HER-2 signalling pathways (Fig. 2d, e). Upstream activator analysis showed that immune infiltrated ROIs were enriched for genes associated with activation of inflammation-associated regulators, including *IL4*, *TNF* and *STAT3* (Fig. 2f).

## Spatial deconvolution of cell types confirmed ROIs represent distinct cell types

As expected, tumour ROIs were highly enriched for malignant and urothelial cells, while stroma ROIs comprised mostly myofibroblast cancer-associated fibroblasts (MyoCAFs) (Fig. 2g). B cells were the most abundant immune population within the stroma, with the proportion of T cells and macrophages varying between stroma ROIs. Other cell types associated with

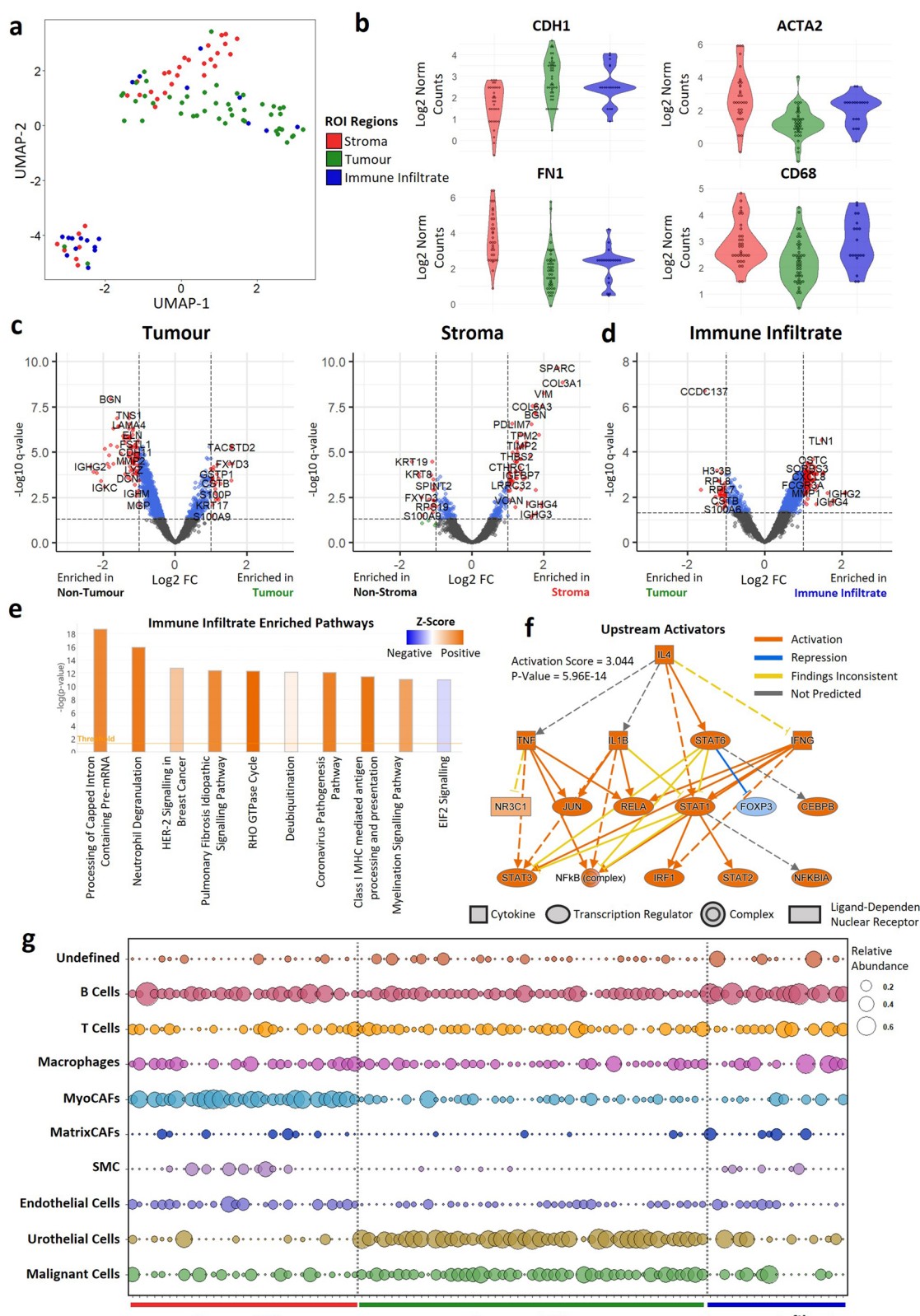

**Fig. 2 | Characterization of tumour, stroma, and immune infiltrate ROIs.**
**a** UMAP dimensional reduction of ROIs showing clustering of tumour (n = 47), stroma (n = 32), and immune infiltrate (n = 19) transcriptomic signatures. **b** Violin plots of epithelial (CDH1), extracellular matrix (ACTA2, FN1) and immune (CD68) marker genes. **c** Volcano plots showing differentially expressed genes between (left) tumour and non-tumour or (right) stroma and non-stroma ROIs. **d** Differentially expressed genes between immune infiltrate and tumour ROIs. **e** IPA analysis of immune infiltrate differentially expressed genes showing enrichment of immune and inflammation associated pathways. **f** IPA upstream activator analysis of immune infiltrate genes. Differentially expressed genes are consistent with activation of a set of inflammation associated regulators including IL4, TNF and STAT3. **g** Spatial deconvolution of cell types in ROIs. Cell transcriptomic profiles were determined using a publicly available single-cell dataset. These cell signatures were then used to predict the proportion of each cell type within each ROI.

the tumour microenvironment (endothelial cells, smooth muscle cells [SMCs], Matrix CAFs) were also enriched in some stroma ROIs. B cells were the dominant cell type in immune infiltrated tumour ROIs, with a subset of ROIs highly enriched with macrophages.

### Good concordance between DSP and bulk sequencing classification of molecular subtypes in tumour ROIs

Molecular subtypes are major drivers of tumour heterogeneity that were identified using bulk sequencing data. Here, tumour ROIs were assigned a mix of basal and luminal subtypes (Fig. 3a) in the expected proportions seen in bulk sequencing datasets[1] (30% basal, 12% LumNS, 24% LumP, 10% LumU), suggesting these signatures originate from tumour regions. The stroma-rich subtype was only found in a single tumour ROI - lower than the expected 15%. However, 60% of stroma ROIs were given the stroma-rich subtype, suggesting this signature is present in most tumours. The remaining 18% of tumour ROIs were not enriched for any subtype signature and classified as undefined. NE-like tumours are rare (1% expected) and were not detected within our dataset. Immune infiltrated tumour ROIs classified as basal or undefined subtypes. Notably, increased immune infiltration is a hallmark of basal tumours.

We compared the molecular subtyping of tumour ROIs with bulk sequencing data performed by Affymetrix Exon Array profiling[6] (n = 28 ROIs, 19 patients) (Fig. 3b). There was 100% concordance between basal subtypes identified using DSP (tumour ROIs) and bulk sequencing. There was 70% concordance in the classification of luminal subtypes in tumour ROIs and bulk sequencing. Stroma-rich tumours identified using bulk sequencing were assigned a mix of basal (25%), luminal (25%) and undefined (50%) subtypes in tumour ROIs.

### DSP identifies key genes and pathways involved in molecular subtypes in situ

We further analysed tumour ROI subtypes by differential gene and IPA analysis. *KRT5* and *GATA3* were among the most highly enriched genes in basal and luminal subtypes respectively (Fig. 3c, d). Basal ROIs were also enriched with neutrophil degranulation, interferon, and HIF1a signalling pathways (Fig. 3e). Upstream activation analysis predicted significant activation of TNF, HIF1A and VEGFA (Supplementary Fig. 2), consistent with findings that Basal tumours are typically more inflamed and hypoxic than other subtypes[1,7]. Luminal ROIs on the other hand were enriched for genes associated with PPARα regulation of lipid metabolism (Fig. 3e). Several genes were also enriched in tumours with an undefined subtype, with the Golgi associated protein *GOLGA6A* being the most significantly enriched (Fig. 3c, d). Undefined ROIs were also enriched for FGF signalling, as well as various interleukin pathways, including IL-17A, IL-23 and IL-13 (Fig. 3e). Single cell spatial deconvolution showed that basal tumours were enriched in T cells, macrophages, and endothelial cells relative to some of the luminal subtypes, while luminal subtypes were enriched in urothelial cells relative to basal tumours (Supplementary Fig. 3).

### Development of a method for using DSP to analyse ligand-signalling interaction in situ

While DSP allows profiling of distinct tumour compartments, the preservation of spatial information within this dataset also enables investigation of the relationships and communication between adjacent regions, such as ligand-receptor interactions. Various ligand-signalling analyses were developed for single-cell RNA-seq datasets. However, lack of spatial information leads to the assumption that any single cell could interact with any other cell. In contrast, we know which tumour regions are adjacent to which stroma regions. With this in mind, we developed a method for ligand signalling analysis using DSP datasets.

Within a tumour, ligands are secreted from one region, which activate receptors in adjacent regions. These receptors then activate signalling pathways and transcription factors, culminating in expression of target genes. Therefore, if a ligand is highly expressed within one region, then its target genes are likely to be highly expressed within the region it is signalling

to (Fig. 4a). Using NicheNet, a ligand-target analysis method developed for single-cell data, we generated a database of all the ligand-target pairs present within the DSP dataset. We then determined the Pearson correlation between all ligands and all their target genes between a sender and receiver region. The mean Pearson correlation between a ligand and its target genes measured ligand activity between regions with r ≥ 0.2 taken as a measure of an actively signalling ligand.

### DSP can analyse active ligand-signalling networks in MIBC

Our method identified for the first time in situ a network of active ligand signalling in bladder cancer. Fifteen ligands signalled from the stroma to activate target genes in the stroma or tumour (Fig. 4b), while 19 ligands originated from tumour regions (Fig. 4c). Gene set enrichment of ligand targets showed that stroma targets were enriched for genes involved in ECM organisation and disassembly (Fig. 4d). These targets included the matrix metalloproteinase *MMP2*, which correlated with stroma derived *THBS2* (r = 0.85), as well as stroma and tumour derived *TNC* (r = 0.76 and 0.64 respectively). Tumour targets were enriched for genes involved in cell proliferation, G1/S transition, and inhibition of apoptosis; for example, *BIRC5* activated by *ITGB1* from the tumour and stroma. These data suggest that tumour targeting ligands promote cell growth and survival, while stroma targeting ligands promote reorganisation and disassembly of the ECM.

To determine the potential origin of active ligands, we correlated ligand expression in each ROI with the cell proportions determined by spatial deconvolution. B cell, endothelial cell and MyoCAF proportions correlated (r = 0.2-0.62) with a range of active ligands (Fig. 4e), suggesting they are likely to be the major source of ligand signalling in MIBC.

### Characterisation of immune infiltrated tumour regions

Immune-infiltrated tumour ROIs showed an increased number of basal signature ROIs (42%) relative to the tumour (28%), and stroma (12%) regions (Fig. 3a). Signatures of inflammation and immune cell trafficking are associated with disease progression[8,9] and high levels of macrophages, amongst other immune suppressive cell types, correlate with a worse outcome in MIBC[10–14]. Therefore, we hypothesised that infiltrating immune cells may secrete ligands that increase the aggressiveness of the tumour. Ligand-target signalling identified nine active ligands signalling from immune infiltrated ROIs to adjacent tumour ROIs (Fig. 5a). The ligand with the highest level of activity was *GPNMB*, a gene highly enriched in basal ROIs (Fig. 3c), which is known to activate the expression of *VEGFA*[15]. *SERPINE1*, another *VEGFA* activator, also showed high ligand activity in immune-infiltrated ROIs. Gene set enrichment identified genes involved in the corticosterone response; a pathway involved in immune reactions and stress response[16], as well as angiogenesis and hypoxia-related genes (Fig. 5b). Immunofluorescence staining confirmed expression of *GPNMB* within the tumour regions in bladder cancer, mirroring the tumour localisation seen in the DSP dataset (Fig. 5c). Additionally, examination of a publicly available bladder tumour single-cell RNA-seq dataset[17] showed high *GPNMB* expression in macrophages, malignant cells and fibroblasts (Supplementary Fig. 4). There were no associations between the expression of *GPNMB* and *VEGFA* and overall survival when analysed using bulk sequencing data from the MIBC patients in the BCON cohort (Supplementary Fig. 5). Finally, we inferred the ligand-to-target signalling path between *GPNMB* and *SERPINE1* to *VEGFA* using the NicheNet package. The most likely signalling route for *SERPINE1* was through the LRP1 receptor, signalling via SRC and SHC1, while *GPNMB* was most likely to signal through EGFR, STAT3 and HIF1A (Fig. 5d).

### Discussion

Our data show that tumour, stroma, and immune infiltrated ROIs of MIBC have distinct transcriptomic profiles in situ. Previous work in bladder cancer using DSP utilized focussed probe panels covering a limited number of RNA or protein targets[3,18]. Thus, we successfully generated, for the first time, a whole transcriptome spatial transcriptomics dataset using bladder cancer

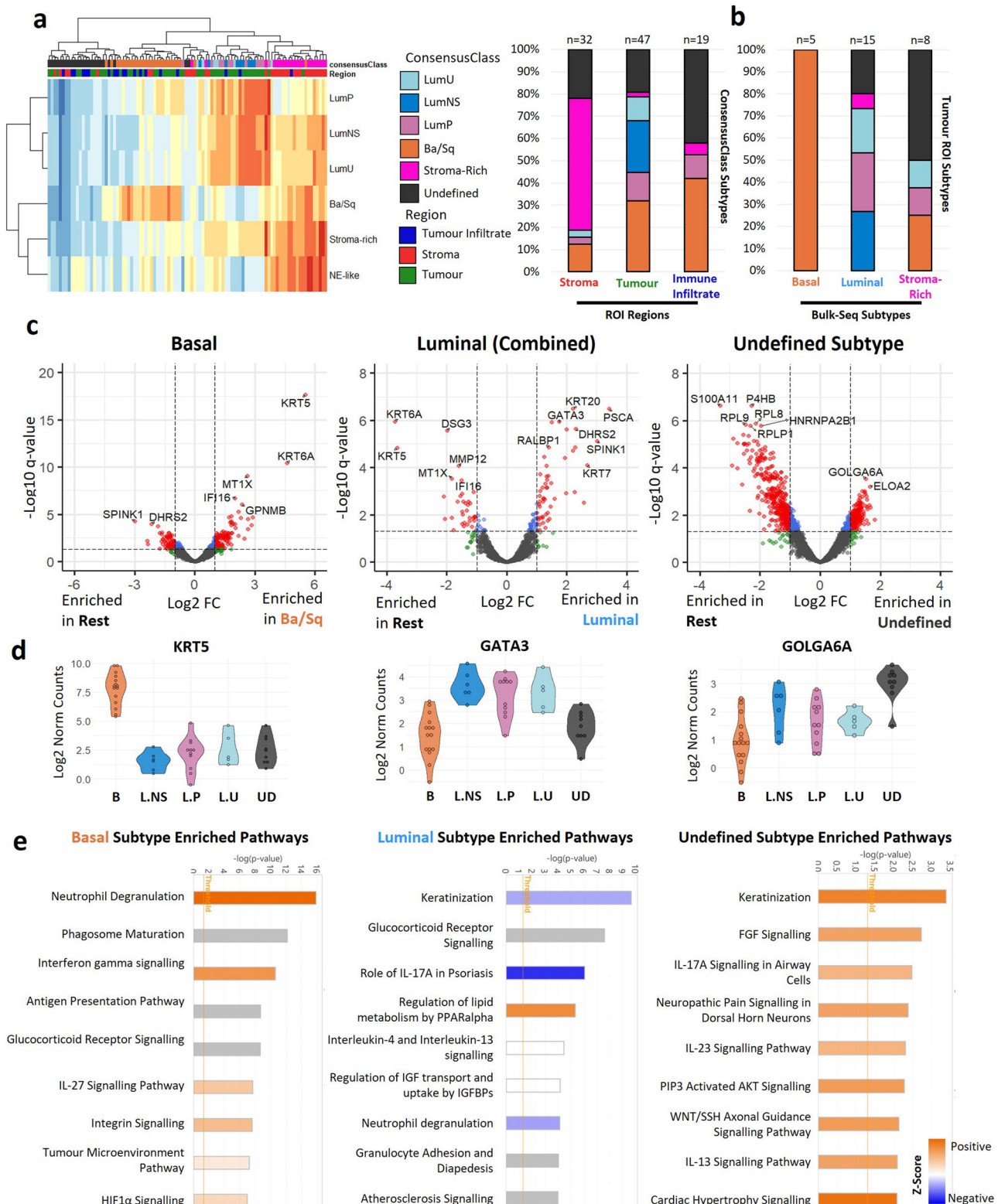

**Fig. 3 | Spatial enrichment of molecular subtype signatures. a** Molecular subtyping of ROIs using the consensusMIBC R package. Left, Heatmap showing enrichment scores for each subtype in tumour, stroma, and immune infiltrated ROIs. Assigned cell types are shown in the top panel. Some ROIs were not enriched for any signature and were assigned an undefined subtype. Right, stacked barplot showing distribution of each subtype in tumour, stroma, and immune infiltrate ROIs. **b** Comparison of molecular subtypes assigned to tumour ROIs and subtypes assigned to the same tumours using conventional bulk sequencing (overlap between datasets n = 28 ROIs, 19 donors). **c** Volcano plots showing differentially expressed genes between basal (n = 15), luminal (n = 22), or undefined (n = 9) tumour ROIs and all other tumour ROIs. **d** Violin plots showing expression of KRT5, GATA3 and GOLGA6A in tumour ROIs. **e** IPA enriched pathways in basal, luminal, and undefined subtype differentially expressed genes.

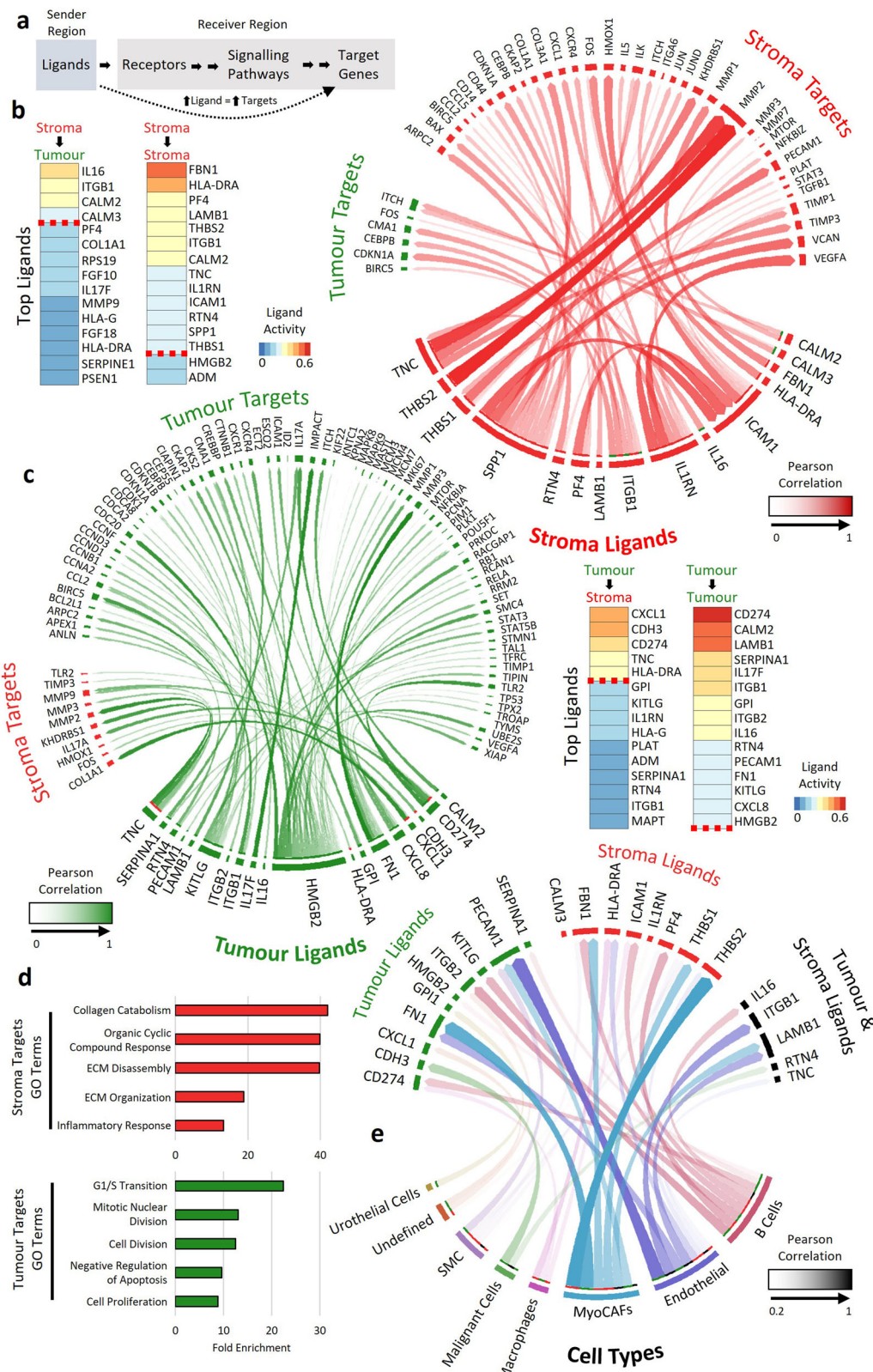

TMAs. Given the age of the tumour samples (16–22 years)[19] the work shows the capabilities and potential of this technology to profile archival tissues and supports further use of stored samples.

Molecular subtyping using bulk sequencing of bladder tumours has revealed a range of subtypes with distinct molecular signatures and clinical outcomes[1]. However, it remains unclear if these signatures originate from tumour or stromal cells, or both. Our results suggest that luminal and basal gene signatures originate from tumour regions, while the stroma-rich signature is present in the stroma of most tumours. This indicates that basal and luminal gene signatures effectively "overwrite" stroma-rich signatures during bulk sequencing analysis, and the stroma-rich subtype is only seen when these other subtypes are not present.

**Fig. 4 | Ligand signalling between tumour and stroma ROIs. a** Overview of ligand to target signalling. Ligands are released from a sender region to activate receptors within a receptor region. These receptors then activate their respective signalling pathways, resulting in the activation of target genes. Therefore, if a ligand is actively signalling to a receiver region, it is expected that there will be a direct correlation between ligand expression in the sender region and target gene expression in the receiver region. The mean Pearson correlation between a ligand and its targets was used to determine a ligands activity between two regions. **b** Left, Top ligands based on ligand activity signalling from stroma to tumour and stroma to stroma ROIs. Right, circos plot showing ligand signalling from stroma ligands to stroma and tumour targets. The opacity of the arrows indicates the Pearson correlation between ligand and target. **c** Right, Top ligands signalling from tumour ROIs to stroma or tumour ROIs. Left, Ligand signalling from tumour ligands to tumour and stroma targets. **d** Gene set enrichment of ligand targets identified in tumour and stroma ROIs. Stroma targeting ligands were enriched for ECM and collagen catabolism associated targets, while tumour targeting ligands were enriched for targets involved in cell survival and inhibition of apoptosis. **e** Circos plot showing links between cell types present in ROIs and the ligands present in that region. Opacity of the arrows indicates the correlation between ligand and cell type. Only correlations above 0.2 are shown.

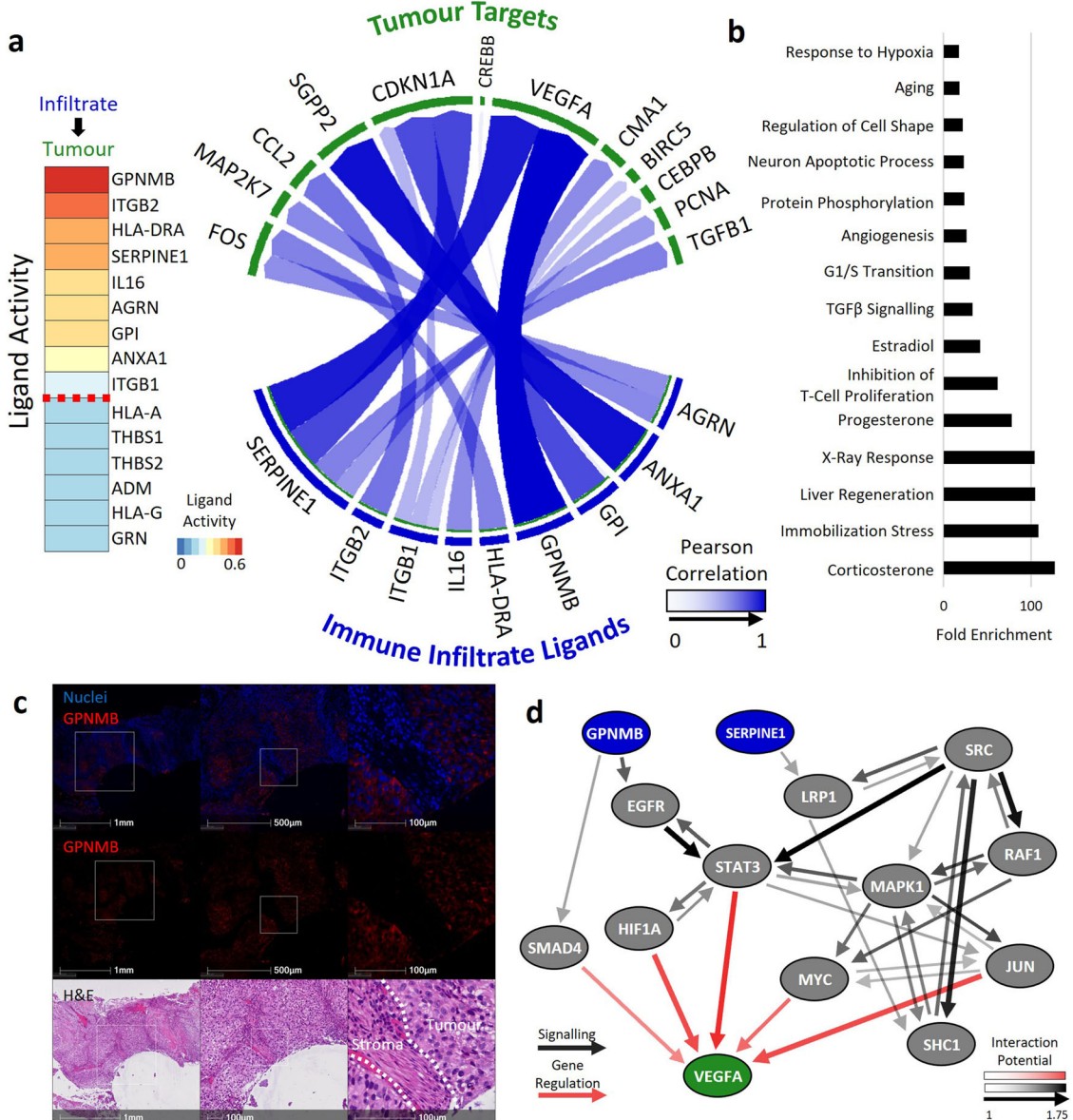

**Fig. 5 | Immune infiltrate ligand signalling events. a** Ligand signalling analysis between immune infiltrate ROIs and adjacent tumour ROIs. Left, top active ligands. Right, Ligands in the immune infiltrate and targets in tumour ROIs. The opacity of the arrows indicates the strength of the correlation between the ligand and the target. **b** Gene set enrichment of immune infiltrate targets. **c** Immunofluorescence staining of one of the top signalling ligands found from Immune infiltrates shows expression of GPNMB in tumour regions. **d** Inferred Ligand-to-target signalling path between GPNMB and SERPINE1 to VEGFA using the NicheNet package. Size and opacity of arrows indicates the interaction potential between two genes.

Several methods have been developed to infer cell-cell communication from scRNA-seq data, such as NicheNet[20] and CellChat[21]. However, as spatial information is lost during scRNA-seq, there is no way to know which cells are physically close to each other, and it is assumed that any cell can communicate with any other cell. We developed a new analysis method for interrogating the communication between adjacent ROIs to study ligand-signalling interactions. This approach builds upon methods developed for single-cell RNA-seq but utilises the fact that the regions in direct contact are known in DSP datasets. This highlighted the potential importance of *GPNMB* signalling prediction, which was shown to be most likely to

signalling through EGFR, STAT3 and HIF1A. This signalling pathway was of particular interest as *GPNMB* was highly enriched in basal tumour ROIs. Notably, immune infiltration, EGFR signalling, STAT3 and hypoxia have all been linked to basal subtypes[22–27], suggesting that GPNMB may play a role in stimulating angiogenesis in basal tumours. GPNMB is a glycoprotein that has been shown to promote the aggressiveness of multiple cancer types, including bladder cancer[28–30]. While we found no significant prognostic evidence when analysing *GPNMB* RNA levels in bulk sequencing, high GPNMB levels measured by immunohistochemistry in bladder cancer was previously correlated with a worse prognosis[29].

One of the main limitations of our study was the low number of samples analysed. However, our finding provide justification for costing a larger study, which will allow us to analyse in situ signalling processing in relation to survival outcomes. While single-cell and bulk sequencing has greatly enhanced our understanding of the molecular underpinnings of bladder cancer, it is limited in its ability to investigate individual sub-compartments within the tumour or signalling events between different regions. A larger spatial dataset may help find new drivers of disease. Another limitation of our study was a lack of tissue available for follow-up experiments to confirm the interaction between GPNMB and VEGFA in bladder cancer. However, this interaction has been well-established elsewhere[15].

In summary, we have generated the first whole transcriptome, spatial dataset of bladder cancer with distinct measurements of tumour, stroma and immune infiltrate regions. This has provided insight into the molecular subtypes underpinning the disease, and newly developed analysis methods such as ligand signalling have helped further our understanding of bladder cancer progression.

## Materials and methods
### Samples
Our retrospective study utilised TMAs generated from pre-treatment MIBC tumours. Patients were recruited into the Bladder Carbogen and Nicotinamide (BCON) trial and randomised between November 2000 and April 2006[19]. A local research ethics committee approved use of BCON samples for translational research (LREC 09/H1013/24). Samples were obtained via transurethral resection prior to formalin-fixation and paraffin embedding. Tumour areas were demarcated by a histopathologist and 1 mm diameter cores (up to three per tumour region from two regions) were taken. TMAs were constructed as described previously[31,32]. Supplementary Table 1 summarises the clinical characteristics of the cohort.

### Sample processing and ROI selection
We performed whole transcriptome DSP of TMAs using the Nanostring GeoMx platform (Seattle, USA) (Fig. 1a). This method utilizes photo-cleavable barcodes attached to RNA probes that are exposed to light within selected regions of interest (ROIs) to generate gene counts, providing homogenised expression data within distinct spatial regions. TMAs were stained with Pan-CK and CD45 using GeoMx Solid Tumour TME Morphology Kits (Nanostring, GMX-RNA-MORPH-HST-12, 1:40 dilution) to highlight tumour and immune cells respectively (Fig. 1b). 300 μm circles were selected within cores at areas with approximately 50% tumour content. These were then segmented into tumour (Pan-CK+ve) and stroma (Pan-CK-ve) ROIs. In some tumours, infiltrating immune cells (Pan-CK+ve and CD45+ve) were also identified (Fig. 1c). There were 137 ROIs selected across 66 cores from 34 patients.

### Library preparation
Sequencing library amplification and cleanup were carried out according to the manufacturer's protocol. Libraries were sequenced by Illumina NovaSeq SP (2x 50 bp). After the initial sequencing, a small proportion of ROIs dominated the reads - attributed to variable tissue quality. An additional sequencing library was then created from the ROIs with low sequencing reads to counteract this.

### Data processing and quality control (QC)
FASTQ files were processed by GeoMx NGS Pipeline (Nanostring, MAN-10118-03) before undergoing QC for minimum raw reads (1000), sequencing saturation ( > 50%) and percent aligned, stitched, and trimmed reads ( > 80%). ROIs with less than 10% of genes expressed were also removed. Following QC, 39 ROIs were excluded, leaving a total of 98 ROIs covering tumour ($n = 47$ ROIs across 28 donors), stroma ($n = 32$ ROIs across 22 donors) and immune infiltrated tumour ($n = 19$ ROIs across 16 donors). Genes undetected in ≤10% of ROIs were removed, leaving 6320 genes. Raw counts then underwent Q3 normalisation.

### Immunofluorescence
Following DSP, tissue sections were stained using haematoxylin 7211 (Fisher Scientific, Hampton, United States, #10034813) and counter-stained with eosin Y (Fisher Scientific, #12677756). H&E-stained slides were stripped using 70% IDA overnight followed by 2% acetic acid overnight. The slides were stained using the automated Leica Bond Rx system (Wetzlar, Germany) using a GPNMB antibody (clone SP299; Abcam, Cambridge, UK, ab227695, 1:100 dilution) before counter-staining with DAPI (ThermoFisher) and mounting with ProLong Gold Antifade Moutant (Thermofisher). Slides were imaged using an Olympus VS120-L100-W-12 microscope (Olympus Corporation, Tokyo, Japan).

### Statistics and reproducibility
Statistical analysis was carried out using R Studio (R version 4.0.3) and graphics produced using ggplot2 unless otherwise stated. Each ROI was treated as an independent replicate and an adjusted p-value threshold of 0.05 was used for all statistical tests. Heatmaps were generated using the pheatmap R package. The R package M3C was used to generate Uniform Manifold Approximation and Projection for Dimensional Reduction (UMAP).

### Differentially expressed genes and gene sets
Genes differentially expressed between ROIs were identified using limma[33]. Gene set enrichment and upstream activator analysis of differentially expressed genes was performed using QIAGEN IPA[34] (QIAGEN Inc.). Gene set enrichment of gene lists was performed using DAVID v6.8[35].

### Cell profile matrix and spatial deconvolution
The cell profile matrix was generated using a single cell RNA sequencing dataset generated by Chung et al.[17]. Additional analysis of the dataset was performed by the Cancer Single-Cell Expression Map[36]. Spatial deconvolution was performed using the SpatialDecon R package[37].

### Ligand-receptor analyses
Active ligand signalling events were predicted by examining the Pearson correlation between ligands and their target genes between sender and receiver regions. A database of ligand-target signalling pairs generated by the R package NicheNet was used[20].

### Molecular subtyping and survival analyses
Molecular subtyping of ROIs was performed using the consensusMIBC package[1]. Six molecular subtypes were identified: Luminal Papillary (LumP), Luminal Nonspecific (LumNS), Luminal Unstable (LumU), Basal/Squamous (Basal), Stroma-rich and Neuroendocrine-like (NE-Like).

Relationships with overall survival and local progression-free survival were assessed via Cox proportional hazard models and Kaplan-Meier curves using survival and survminer packages in R using 16-year follow up data[27].

### Reporting summary
Further information on research design is available in the Nature Portfolio Reporting Summary linked to this article.

## Data availability

All data required to replicate the analysis carried out in this paper is present in the supplementary data. This includes raw, normalised and log2 counts, ROI annotations, differentially expressed gene results for ROI regions and subtypes, and the cell profile matrix used for cell type deconvolution.

## Code availability

R analysis script containing most statistical methods, including ligand analysis, is available in the supplementary files.

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

## Acknowledgements

Digital spatial profiling was funded by Innovate UK. Ananya Choudhury, Peter Hoskin and Catharine West are supported by the National Institute of Health Research (NIHR) Manchester Biomedical Research Centre, UK (NIHR129943). The work was also supported by Cancer Research UK (CRUK) funding for the Manchester Radiotherapy Research Centre of Excellence (RadNet C19942/A28832), the Manchester Major centre (CTRQQR-2021\100010), and project grants (C1098/A9437; C2094/A11365). Immunofluorescence staining was performed by Histology in the Cancer Research UK Manchester Institute.

## Author contributions

Michael Eyers, conceptualisation, methodology, software, writing—original draft, Joely Irlam, conceptualisation, Gayle Marshall, conceptualisation, Vicky Smith, Validation, Investigation, Alexander Baker, Investigation, Lucy Frost, Investigation, Peter Hoskin, Resources, Ananya Choudhury, Resources, Catharine West, Conceptualisation, Writing—Review & Editing

## Competing interests

The authors declare no competing interests.
