## [Peer Review File · Communications Biology]

Reviewers' comments:

Reviewer #1 (Remarks to the Author):

This manuscript conducted whole transcriptome DSP on bladder cancer with tumor, stroma and immune infiltrated tumor regions and explored the spatial relationship of molecular subtype signatures and ligand-receptor signalling events. This research contains novelty and innovation, but there are the following deficiencies:

1. The raw data must first provide a link that reviewers can access and the data must be made public before publication;
2. An important basis for reliable conclusions of scientific research is its reproducibility. The lack of source data or raw data and the messy analysis scripts lead to poor reproducibility, and the author needs to re-organize and upload the corresponding files;
3. Please unify "tumor infiltrated immune" and "immune infiltrated tumor" in the manuscript and figures.
4. The resolution of the manuscript pictures is insufficient, which makes it difficult to read. Please provide drawings with sufficient resolution;
5. In Fig2A, please provide immune cell markers plots.
6. Line 167 and 171, it should be Fig3D instead of Fig3C. Fig3C is not described in the manuscript.
7. There is no Fig4D in Fig4.

Reviewer #2 (Remarks to the Author):

The manuscript "Digital Spatial Profiling of the Microenvironment of Muscle Invasive Bladder Cancer." is exciting and holds significant potential in the field. However, I would like to suggest some areas of improvement to enhance the clarity and impact of your paper:

1. The term "BCON" is frequently mentioned in the materials and methods section but is unclear. Please provide a definition or full form for BCON. Additionally, have you considered using tools like CellChat for the ligand-receptor analyses? It could give more insights.
2. Detailed Descriptions for Figure 2: There are a few aspects of Figure 2 that would benefit from further explanation:
 - A. For "Figure A," please provide more details about the axes, particularly what dimensions 1 and 2 represent.
 - B. Clarify whether the data is derived from single-cell or bulk RNA-seq. Indicate the number of cells or samples analyzed for each condition.
 - C. "Figure C" needs a more detailed explanation of the x and y axes and the observed patterns or clusters.
 - D. The gene set enrichment analysis in "Figure C" could be more informative if focused on differentially expressed genes and their association with specific KEGG pathways in different conditions or clusters.
3. Providing a more straightforward narrative or summary that connects the analyses presented in Figures 2 and 3 would be beneficial. Currently, the relationship between these figures and the overall story of the paper is not immediately apparent.
4. Enhancing the figures' quality and captions would significantly improve the visual communication of findings. While the analyses conducted are extensive, their connection to the overall narrative of the paper could be strengthened for better coherence and impact.

5. The author mentions the limited number of samples in the discussion section. Discussing how a larger sample size might improve future analyses would be insightful. Elaborating on the new analysis method mentioned in the discussion section, especially regarding how it interrogates the communication between adjacent ROIs for studying ligand-signaling interactions.
6. Demonstrating biological relevance through differential expression analysis between conditions for each subtype would significantly enhance the value of findings.
7. Improving the clarity of legends in some figures and their direct connection to the results section would aid reader comprehension.
8. Overall, the manuscript would benefit from revisions for clarity and conciseness, particularly in the abstract, methods, and results sections.

Reviewer #3 (Remarks to the Author):

The authors should add a table with the clinical and histopathological details related to the patients used in this analyses. Furthermore details on the selection of the ROIs should be provided, the ROI's in figure 1 shows mixtures of different cell types. If the aim was to differentiate between the different cell types why did the authors not select pure tumor or stroma ROI's? How were these ROIs selected any criteria that the tumor ROIs should have more than 50% tumor cells?

The only description of the TMAs in the references is the following "up to 3 per tumour region from 2 regions" but were the cores taken in certain area's? Close to the invasive margins, close to the hypoxic areas? How reproducible were the ROIs from the same patients? Were these ROIs added in the total or was the average of two ROI's from the same patient added?

If the ROI's contained both TILs and tumor cells were the signals normalized towards the contents? Was there any difference in DGE between the good quality ROIs and the bad quality ROIs? What cut off did the authors use to separate these two groups from one another?

When presenting the result from the different analyses like for example the subtypes, the authors should add the number of ROIs that were included in the analyses.

Using the spatial information was there any heterogeneity within the tumor regions investigated Was there a different expression of the tumor cells close to the TILs vs those further away from the TILs? Same for the TILs?

Related to the ligand analyses, was the ligand and target region calculated by taken the whole tumor vs the whole stroma region in one ROI? Could the analyses be restricted to fewer cell layers? If the whole region is taken have the authors looked at a dilution or diffusion effect of these target genes? The immunofluorescence is nice but then the same should be done for VEGFA to really show that the hypothesis about ligands and targets is correct.

Response to Reviewers

We thank the reviewers for their input and their help in improving our manuscript. We have implemented most of their suggestions and have highlighted areas of the manuscript with major changes in red. Each of the reviewer comments have been addressed below:

Reviewer #1 (Remarks to the Author):

This manuscript conducted whole transcriptome DSP on bladder cancer with tumor, stroma and immune infiltrated tumor regions and explored the spatial relationship of molecular subtype signatures and ligand-receptor signalling events. This research certains novelty and innovation, but there are the following deficiencies:

1.The raw data must first provide a link that reviewers can access and the data must be made public before publication;

Raw and normalised data are now available in the supplementary data.

2.An important basis for reliable conclusions of scientific research is its reproducibility. The lack of source data or raw data and the messy analysis scripts lead to poor reproducibility, and the author needs to re-organize and upload the corresponding files;

We have now included all of the relevant materials to replicate the analysis in this study in the supplementary files. This includes: Raw and normalised counts, annotations file, R Script containing all statistical tests carried out, full results from differential expression and IPA analysis, and the cell profile matrix used for spatial deconvolution of cell types.

3.Please unify “tumor infiltrated immune” and ”immune infiltrated tumor” in the manuscript and figures.

The text is now unified as suggested.

4.The resolution of the manuscript pictures is insufficient, which makes it difficult to read. Please provide drawings with sufficient resolution;

We increased the resolution of figures that were difficult to read and reuploaded them at a larger size.

5.In Fig2A, please provide immune cell markers plots.

We have added CD68 to these plots. Note that we have also changed these plots and the plots in figure 3D to show Log2 normalised counts rather than just normalised counts.

6.Line 167 and 171,it should be Fig3D instead of Fig3C.Fig3C is not described in the manuscript.

This figure has now been reworked in line with the suggestions from reviewer 2 and the text has been rewritten. Figure 3D now shows IPA pathway enrichment for genes enriched in each subtype.

7. There is no Fig4D in Fig4.

This has now been corrected.

Reviewer #2 (Remarks to the Author):

The manuscript "Digital Spatial Profiling of the Microenvironment of Muscle Invasive Bladder Cancer." is exciting and holds significant potential in the field. However, I would like to suggest some areas of improvement to enhance the clarity and impact of your paper:

1. The term "BCON" is frequently mentioned in the materials and methods section but is unclear. Please provide a definition or full form for BCON. Additionally, have you considered using tools like CellChat for the ligand-receptor analyses? It could give more insights.

We added a definition for BCON in the methods section of the paper.

We have looked at both NicheNet and CellChat. However, both methods require an input list of genes, then look for overexpression of potential ligand that could be causing this overexpression. Furthermore, as they have been developed for single-cell data, both methods look at two populations of cells (eg by taking the average expression of each ligand), without any consideration for whether those cells are actually close to each other. Finally, the number of cells in single-cell experiments is far greater than the number of ROIs in a typical DSP experiment, meaning that methods for single-cell experiments may be underpowered when working with DSP data. This is what drove us to develop a new method.

The main advantages of our method are that it considers the spatial aspect of the data by looking at highly correlated ligands and targets in adjacent regions. It also does not require an input list of genes, as it picks up all highly correlated ligands.

We did use the Ligand-Target database generated by NicheNet in order to develop our method. CellChat have similarly created a manually curated signalling interaction database that takes into account the structural composition of ligand-receptor interactions. As our method only considers the downstream transcriptional targets, and not the receptor itself, CellChat could be used to help narrow down potential hits.

We also note that a recent preprint paper by Jin et al adds a spatial element to the CellChat methodology to inspect spatially proximal cell-cell communications (<https://www.biorxiv.org/content/10.1101/2023.11.05.565674v1>). However, our method is still distinct from this as this method is used on spot data (10X visium, SlideSeq etc), whereas the DSP datasets consist of discrete regions within the tissue that are adjacent to one another

2. Detailed Descriptions for Figure 2: There are a few aspects of Figure 2 that would benefit from further explanation:

A. For "Figure A," please provide more details about the axes, particularly what dimensions 1 and 2 represent.

The axes on this plot have been changed to UMAP-1 and UMAP-2 to make it clearer that this is a UMAP dimensional reduction. We have also added a

definition for UMAP in the Materials and methods under the Data Analysis section.

B. Clarify whether the data is derived from single-cell or bulk RNA-seq. Indicate the number of cells or samples analyzed for each condition.

All data from this figure is derived from the DSP dataset. The cell type deconvolution in figure 2G uses a single-cell RNA-Seq dataset to create a cell profile matrix which is then applied to the DSP data to give the proportion of cells per ROI. We have modified the figure legend to make this clearer. We have also added the number of samples for stroma, tumour, and immune infiltrate in the figure legend. Additionally, we have included the cell profile matrix used for cell deconvolution in the supplementary files.

C. "Figure C" needs a more detailed explanation of the x and y axes and the observed patterns or clusters.

This figure has now been reworked as suggested in D.

D. The gene set enrichment analysis in "Figure C" could be more informative if focused on differentially expressed genes and their association with specific KEGG pathways in different conditions or clusters.

This figure has been reworked and now shows differentially expressed genes between ROIs (tumour vs non-tumour, stroma vs non-stroma (Figure 2C) and immune infiltrate vs tumour (figure 2D). IPA analysis was then carried out on differentially expressed genes for each type of ROI. Tumour and stroma results are shown in supplementary figure X and immune infiltrate results are shown in figure 2E, along with upstream activator analysis (Figure 2F). Full results from pathway analysis are now available in supplementary data.

3. Providing a more straightforward narrative or summary that connects the analyses presented in Figures 2 and 3 would be beneficial. Currently, the relationship between these figures and the overall story of the paper is not immediately apparent.

We have modified the results, discussion, and figure legends throughout to improve coherence and better tie the story together. This includes gene set enrichment for immune infiltrate ROIs in figure 2 showing enrichment of inflammation associated genes. Figure 3 now shows gene set enrichment for Basal tumours showing enrichment of neutrophil degranulation genes, and upstream activator analysis of Basal tumours also show enrichment of HIF1A, VEGFA and other inflammation associated factors. GPNMB can also be seen as a highly enriched Basal associated gene in figure 3C. This helps to tie the concept of GPNMB promoting tumour aggressiveness in Basal tumours by promoting VEGFA, via HIF1A and STAT3, as mentioned in the discussion.

4. Enhancing the figures' quality and captions would significantly improve the visual communication of findings. While the analyses conducted are extensive, their connection to the overall narrative of the paper could be strengthened for better coherence and impact.

We have modified the results, discussion, and figure legends to improve coherence and better tie the story together. We have also reuploaded figures at a higher resolution

5. The author mentions the limited number of samples in the discussion section. Discussing how a larger sample size might improve future analyses would be insightful. Elaborating on the new analysis method mentioned in the discussion section, especially regarding how it interrogates the communication between adjacent ROIs for studying ligand-signaling interactions.

We have expanded this section in the discussion.

6. Demonstrating biological relevance through differential expression analysis between conditions for each subtype would significantly enhance the value of finding

As suggested, Figure 3E has been reworked in a similar manner to Figure 2C to show gene sets significantly enriched in each subtype (Figure 3E and Supplementary) and the results section has been modified accordingly. The new analysis showed enrichment of many preestablished pathways in the basal subtype, as well as a role for interleukin signalling in the undefined subtype. Note that the volcano plots in figure 3C are now slightly different as previously the single stroma-rich subtype tumour ROI was excluded. However, this was not necessary as we are comparing one subtype to all other subtypes. We have also moved the cell deconvolution results to the supplementary.

7. Improving the clarity of legends in some figures and their direct connection to the results section would aid reader comprehension.

As suggested, the figure legends have been reworked to improve clarity and connection to the results.

8. Overall, the manuscript would benefit from revisions for clarity and conciseness, particularly in the abstract, methods, and results sections.

We have modified the text and figure legends to improve comprehension.

Reviewer #3 (Remarks to the Author):

The authors should add a table with the clinical and histopathological details related to the patients used in this analyses. Furthermore details on the selection of the ROIs should be provided, the ROI's in figure 1 shows mixtures of different cell types. If the aim was to differentiate between the different cell types why did the authors not select pure tumor or stroma ROI's? How were these ROIs selected any criteria that the tumor ROIs should have more than 50% tumor cells?

We have added a table showing clinical and histopathological details of the cohort used. We have also updated the methods to better explain the ROI selection.

The only description of the TMAs in the references is the following "up to 3 per tumour region from 2 regions" but were the cores taken in certain area's? Close to the invasive margins, close to the hypoxic areas? How reproducible were the ROIs from the same patients? Were these ROIs added in the total or was the average of two ROI's from the same patient added?

If the ROI's contained both TILs and tumor cells were the signals normalized towards the contents?

The methods section has been updated to include how tumour cores were taken. We have also added a supplementary figure showing hierarchical clustering of the top 1000 variable genes from tumour, stroma, and immune infiltrate ROIs to show how ROIs from the same patient cluster. Some cores from the same patient did cluster together, indicating similar transcriptomic profiles. However, most ROIs did not, suggesting high levels of intra-tumour heterogeneity. Therefore, all ROIs were included in analysis rather than taking the average of multiple ROIs from the same patient.

Was there any difference in DGE between the good quality ROIs and the bad quality ROIs? What cut off did the authors use to separate these two groups from one another?

Bad quality ROIs were removed during the QC steps so were not included in any further analysis. The following criteria were used to remove poor quality ROIs: >50% sequencing saturation, >1000 raw reads, >80% trimmed, stitched and aligned reads, and more than 10% of genes expressed. Additional text has been added to the methods section under Data Processing and Quality Control to better explain this.

When presenting the result from the different analyses like for example the subtypes, the authors should add the number of ROIs that were included in the analyses.

We have added this into the figures and figure legends.

Using the spatial information was there any heterogeneity within the tumor regions investigated Was there a different expression of the tumor cells close to the TILs vs those further away from the TILs? Same for the TILs?

There were no significant differences between tumour ROIs close to/far from TILs. We have added supplementary figure 6 which shows the hierarchical clustering of the top variable tumour, stroma and immune infiltrate genes. There are multiple distinct clusters in each type of ROI suggesting significant heterogeneity. However, this was not explored further in this paper.

Related to the ligands analyses, was the ligand and target region calculated by taken the whole tumor vs the whole stroma region in one ROI? Could the analyses be restricted to fewer cell layers? If the whole region is taken have the authors looked at a dilution or diffusion effect of these target genes?

For the ligand signalling we used the tumour and stroma ROIs within the same core. DSP provides RNA counts by exposing a selecting area to UV light to cleave photocleavable barcodes. Therefore, each ROI actually represents the mean expression of genes within that area and there is no way to further restrict cell layers within ROIs after generating the dataset. Additional text has been added to the methods section under Sample Processing and ROI selection.

GPNMB can signal in a paracrine fashion through shedding of its extracellular domain. However, to my knowledge it's not clear how far this signalling can reach.

The immunofluorescence is nice but then the same should be done for VEGFA to really show that the hypothesis about ligands and targets is correct.

Unfortunately, the tissue blocks used in this project have been exhausted so there is no tissue available for us to perform additional staining (the GPNMB IF was carried out by stripping a H&E section for this reason). However, the interaction between GPNMB and VEGFA has been well established elsewhere (Taya and Hammes, 2018). Additional text has been added into the discussion under the limitations of the study section to highlight this point.

REVIEWERS' COMMENTS:

Reviewer #1 (Remarks to the Author):

Most of the deficiencies have been solved, but some others remain:

1. Some abbreviations were not given full names when they were first mentioned, such as MyoCAFs, etc.
2. In Fig2D-E, why do the authors conduct differential analysis between immune-infiltrate and tumour instead of immune-infiltrate and non-immune-infiltrated?
3. In Fig2F and SFig2, please indicate what the shape and color of the gene represents.
4. In Fig2F, please add the Activation Z-Score and P-Value.
5. The resolution of most images is still unsatisfactory.

Reviewer #3 (Remarks to the Author):

The answers and changes are satisfactory.

Response to Reviewers

We thank the reviewers for their input and their help in improving our manuscript. We have implemented their suggestions, and the comments are addressed individually below:

Reviewer #1 (Remarks to the Author):

1. 1. Some abbreviations were not given full names when they were first mentioned, such as MyoCAFs, etc.
 - We have added in full names for any abbreviations that were missing.

2. 2. In Fig2D-E, why do the authors conduct differential analysis between immune-infiltrate and tumour instead of immune-infiltrate and non-immune-infiltrated?
 - As the immune infiltrate ROIs consist of segmented tumour ROIs (i.e. immune cells within the tumour regions), we reasoned that comparing the immune infiltrate to the tumour ROIs would give a clearer indication of immune infiltrate specific genes.

3. 3. In Fig2F and SFig2, please indicate what the shape and color of the gene represents.
 - We have added a legend at the bottom of the figure indicating what the shape of each gene represents. There is a legend to the right of the figure indicating what the different colours indicate.

4. 4. In Fig2F, please add the Activation Z-Score and P-Value.
 - This has now been added.

5. 5. The resolution of most images is still unsatisfactory.
 - We hope that the final upload of the figures will resolve any resolution issues.

Reviewer #3 (Remarks to the Author):

The answers and changes are satisfactory.